# Diagnostic Value of Middle Meatal Cultures versus Maxillary Sinus Culture in Acute and Chronic Sinusitis: A Systematic Review and Meta-Analysis

**DOI:** 10.3390/jcm11206069

**Published:** 2022-10-14

**Authors:** Do Hyun Kim, Sung Won Kim, Mohammed Abdullah Basurrah, Se Hwan Hwang

**Affiliations:** 1Department of Otolaryngology—Head and Neck Surgery, Seoul Saint Mary’s Hospital, College of Medicine, The Catholic University of Korea, Seoul 06591, Korea; 2Department of Surgery, College of Medicine, Taif University, Taif 21944, Saudi Arabia; 3Department of Otolaryngology—Head and Neck Surgery, Bucheon Saint Mary’s Hospital, College of Medicine, The Catholic University of Korea, Bucheon 14647, Korea

**Keywords:** maxillary sinus, puncture, nasal cavity, sinusitis, meta-analysis

## Abstract

Background: To assess the diagnostic utility of middle meatal culture (MMC) in patients with acute and chronic sinusitis; Methods: Six databases were thoroughly reviewed up to March 2022. Sensitivity, specificity, and negative and positive predictive values were extracted. Methodological quality was evaluated using the QUADAS-2 instrument; Results: Fifteen reports were analyzed. MMC results exhibited a significant correlation (r = 0.7590, 95% confidence interval [CI] [0.6855; 0.8172], *p* < 0.0001) with those of maxillary sinus puncture. The diagnostic odds ratio (DOR) of MMC (reference = maxillary sinus culture) was 8.5475 [3.9238; 18.6199]. The area under the summary receiver operating characteristic curve was 0.761. The sensitivity and specificity of MMC were 0.7759 [0.6744; 0.8526] and 0.7514 [0.6110; 0.8534], respectively. We performed subgroup analysis based on age (children vs. adults), duration of disease (acute vs. chronic), and specimen collection method (biopsy, swabs, suction tips). The DORs, specificities, and negative and positive predictive values varied significantly. Diagnostic accuracy was highest for children and individuals with chronic disease, and when samples were collected via suction.; Conclusions: MMC provided fair diagnostic accuracy in patients with acute or chronic sinusitis. Although some institutional differences were evident, the middle meatal and maxillary sinus culture results were similar.

## 1. Introduction

Rhinosinusitis is one of the most common primary care diagnoses and the fifth most common reason for antibiotic prescription [1] due to resistance to first-line antibiotics [2]. Treatment should be more precise in the present era of increasing bacterial antimicrobial resistance. Inadequate treatment not only increases medical costs, but may also lead to recurrent or chronic rhinosinusitis [3]. Culture-directed therapy is known as the optimal treatment option [4]. Maxillary sinus culture using fluid directly aspirated through the canine fossa has been considered as the gold standard [2,5]. However, puncture is invasive, must be performed under local or general anesthesia, and can cause discomfort, pain, and inflammation [1]. After the introduction of endoscopes, middle meatal culture (MMC) has taken over the need for maxillary sinus puncture [4]. Several studies have assessed the correlation between middle meatal and maxillary sinus cultures, but the data are inconsistent, and the sample sizes have tended to small [2,6,7,8]. Some reviews used statistical analyses to clarify the results [2,9]. However, these articles appeared over 15 years ago, since which new studies have been published; moreover, no previous studies examined pediatric populations [4,10,11,12,13]. Therefore, we performed a meta-analysis to explore the relationship between MMC and maxillary sinus aspiration (MSA) (followed by culture). We performed subgroup analyses by age (children vs. adults), disease duration (acute vs. chronic), and specimen collection methods (biopsy vs. swab vs. suction).

## 2. Materials and Methods

### 2.1. Study Registration and Search Strategy

The study protocol was prospectively registered on the Open Science Framework (Charlottesville, VA, USA) (https://osf.io/3v9tu/, accessed on 29 March 2022). We searched PubMed, Embase, the Web of Science, the Cochrane library, SCOPUS, and Google Scholar from the dates of their inception to March 2022. The Population, Intervention, Comparison, and Outcome (PICO) parameters were as follows: P, patients who underwent MMC; I, bacterial growth assessed in the MMC; C, culture of maxillary sinus specimens; and O, bacterial growth. Only papers written in English were considered. The search terms were “bacteriology”, “microbiology”, “sinus”, “microbiology”, “antral”, “antral puncture”, “maxillary tap”, “sinus tap”, “culture”, “middle meatus”, “middle meatal”, “sinusitis”, “acute sinusitis”, “chronic sinusitis” and “maxillary.” Two independent reviewers (SWK and DHK) systematically reviewed the titles and abstracts of all candidate studies. Any disagreements were discussed with a third reviewer (SHH). We checked all reference lists to ensure that no relevant studies were missed.

### 2.2. Selection Criteria

The inclusion criteria were as follows: bacteriological analysis of MMC; cohort study; and comparison between MMC and maxillary sinus culture (with the latter serving as the reference). The exclusion criteria were as follows case reports or reviews; diagnosis using a non-MMC (e.g., a nasopharyngeal swab); and/or insufficient data for calculation of sensitivity and specificity. A flow chart of the search strategy is shown in Figure 1.

### 2.3. Analysis Methods and Bias Assessment

Study data were collected using a standardized form [14,15]. We evaluated the diagnostic power of MMC bacteriology (i.e., the diagnostic odds ratio [DOR] [16]), and then drew summary receiver operating characteristic (ROC) curves and measured the areas under the curve (AUC) [1,3,4,6,7,8,9,10,11,12,13,17,18,19,20]. True-positive, true-negative, false-positive, and false-negative data were collected to calculate the AUCs and DOR (as a single indicator of diagnostic performance). A DOR of 1 indicates that it is not known whether a disease is present. The inverse of fractions from 0 to 1 is calculated; the larger the value, the better the diagnostic performance. DOR values are presented with 95% confidence intervals (CIs). Summary ROC analysis is the most common analytical method in studies that consider sensitivity and specificity. AUC values range from 0 to 1; the higher the value, the higher the diagnostic power (value of 1 indicates 100% sensitivity and specificity). We used random-effects models because within- and between-study variations were apparent.

Using maxillary sinus aspiration (MSA) (followed by culture) as the reference, the diagnostic outcome of MMC in terms of bacterial detection was considered true-positive when both the MMC and MSA cultures were positive and shared at least one pathogen, true-negative when no growth was evident in either culture, false-positive when the MMC culture was positive but that of MSA was negative; and false-negative when the MMC culture was negative or both cultures were positive but the bacteria differed [8]. Using maxillary sinus aspiration (MSA) (followed by culture) as the reference, the diagnostic outcome of MMC in terms of bacterial detection was considered true-positive when both the MMC and MSA cultures were positive and shared at least one pathogen, true-negative when no growth was evident in either culture, false-positive when the MMC culture was positive but that of MSA was negative; and false-negative when the MMC culture was negative or both cultures were positive but the bacteria differed [2]. Study quality was assessed using the Quality Assessment of Diagnostic Accuracy Studies ver. 2 (QUADAS-2) tool [21].

### 2.4. Statistical Analysis and Outcomes

R software (version 3.6.3; R Foundation, Vienna, Austria) was used for the meta-analysis. The Q statistic was used to analyze heterogeneity. Forest plots were drawn to visualize correlations among MMC and MSA culture data, DORs, and sensitivity, and specificity values. Summary ROC curves were drawn, and subgroup analyses were performed by age (children vs. adults), disease duration (acute vs. chronic), and sampling method (biopsy vs. swab vs. suction). The pooled correlation coefficient between the MMC and MSA measurements was calculated. We used Fisher’s *r* to obtain approximately normally distributed *z* values for calculation of the 95% CIs. The correlations were classified as poor (*r* < 0.20), average (*r* = 0.20–0.39), moderate (*r* = 0.40–0.59), strong (*r* = 0.60–0.79), or very strong (*r* > 0.80) [22]. During sensitivity analysis, we removed each study individually to assess its contribution to the overall effect sizes. We drew Begg funnel plots and used the Egger linear regression test to assess publication bias.

## 3. Results

We included 15 studies with 881 subjects. The study characteristics and bias assessments are shown in Appendix A. The Begg funnel plot (Appendix A) revealed no obvious bias. The Egger test (*p* > 0.05) indicated that the risk of publication bias was low.

### 3.1. Correlations between Middle Meatus and Maxillary Sinus Cultures

The pooled r value was 0.7590 (95% CI [0.6855; 0.8172], *p* < 0.0001, I^2^ = 56.0%; Figure 2), indicating a statistically significant association. Significant inter-study heterogeneity was apparent (I^2^ > 50%), attributable to the inclusion of subjects of different ages (adults vs. children), and to differences in sinusitis duration (acute vs. chronic) and sampling methods (biopsy vs. swab vs. suction). Subgroup analysis revealed no significant difference by age (adults 0.7563 vs. children 0.7849; *p* = 0.6809), disease duration (acute 0.7458 vs. chronic 0.7745; *p* = 0.6637) or sampling method (biopsy 0.7200 vs. swab 0.7115 vs. suction 0.8353; *p* = 0.0547) (Table 1). However, suction collection tended to be more strongly correlated with the maxillary sinus culture data than the MMC data. Compared to swabbing, suction samples showed a more significant correlation with the maxillary sinus culture data (*p* = 0.0227).

### 3.2. Diagnostic Accuracy of Middle Meatus and Maxillary Sinus Cultures

The DOR for MMC compared with MMA was 8.5475 (95% CI [3.9238; 18.6199]; I^2^ = 51.0%; Figure 3). The AUC was 0.761. An area under the summary ROC (SROC) curve of 0.70–0.8 is considered fair (Figure 4) [23]. In this study, the sensitivity and specificity were 0.7759 ([0.6744; 0.8526]; I^2^ = 67.4%) and 0.7514 ([0.6110; 0.8534]; I^2^ = 58.2%), respectively (Figure 5). However, high heterogeneity (I^2^ ≥ 50%) was evident, attributable to the inclusion of various subgroups. We found significant differences in sensitivity (*p* = 0.3152) and the DOR (*p* = 0.1045) between adults and children specificity was higher for children than adults (0.9600 vs. 0.6704; *p* = 0.0193). In terms of disease duration, the sensitivity (0.8219 vs. 0.7184; *p* = 0.2338) and specificity (0.8226 vs. 0.6694; *p* = 0.2256) showed a trend toward being higher for the chronic compared with the acute subgroup, and while the diagnostic accuracy was significantly higher in the former subgroup (18.2397 vs. 4.2660; *p* = 0.0373). In terms of the sampling methods, biopsy showed the highest sensitivity but lowest specificity and diagnostic accuracy. Suction had high sensitivity (0.7979) and diagnostic accuracy (15.1597), and was significantly more specific than biopsy and swabbing (0.8400 vs. 0.2857 and 0.7000, respectively; *p* = 0.0158).

## 4. Discussion

To the best of our knowledge, this is the most recent and largest meta-analysis to perform subgroup analyses by age group, sampling method, and disease duration to assess the diagnostic accuracy of MMC in terms of the bacteriology of sinusitis. MMC had a pooled correlation of 0.7590 with maxillary sinus puncture results, and a pooled sensitivity of 0.7759, pooled specificity of 0.7514, and AUC of 0.761. AUC values in the range 0.70–0.80 indicate moderate diagnostic accuracy [24], As the pooled sensitivity and specificity range from 75% to 79%, MMC does not reliably exclude false-negatives or -positives. Negative test results may be false for ~30% of patients; in such cases, an important pathogen is missed, and the antibiotics required are thus not given. False-positives are associated with overtreatment and the development of antibiotic-resistance. Two previous meta-analyses assessed the efficacy of MMC for detecting the pathogens associated with sinusitis. Benninger at al. (2006) reviewed four studies including 121 patients. Known pathogenic bacteria were detected with a sensitivity of 80.9% and specificity of 90.5%. However, neither the review methodology nor the statistical method used to derive the values was explained [9]. Dubin et al. (2005) reviewed seven studies including 650 acute and chronic sinusitis patients. The bacterial culture sensitivity was ~75% and the specificity was 54~62%. The methodology and statistical analyses were clearly explained and the outcomes, although conservative, were accurate [2]. However, although the authors acknowledged the heterogeneity of the data and outcomes (sensitivity and specificity differed between fixed- and random-effects models), subgroup analysis was not used to identify the relevant factors.

The increased involvement of multi-antibiotic-resistant bacteria in upper respiratory tract infections is of concern, particularly in patients with acute or chronic sinusitis prescribed broad-spectrum antibiotics to treat persistent or recurrent non-specific symptoms [2]. *Streptococcus pneumoniae*, *Hemophilus influenzae*, *Moraxella catarrhalis*, and *Staphylococcus aureus* are the major pathogens of acute bacterial sinusitis [23,24]. In this study, it was important to confirm that MMC (after endoscopic aspiration or swabbing) adequately replaced MAS (with culture). We performed subgroup analyses by age and sampling method to explore the impact of those variables on diagnostic accuracy.

We performed a bivariate meta-analysis using the DORs and the pooled sensitivities and specificities. As the DOR is closely linked to both sensitivity and specificity, we were able to derive odds ratios, in line with standard meta-analytical practice. The DOR is useful for assessing correlations between sensitivity and specificity, and permits evaluation of study heterogeneity by stratifying relevant parameters. Thus, the DOR is a robust meta-analytical tool.

We found that patient age, disease duration, and sampling type significantly affected diagnostic accuracy. The nasal passage of children is significantly narrower than that of adults; therefore, contamination of endoscopically obtained specimens might be more common in the former group [25]. This may explain why the AUC (0.7079) for children in this study was lower than that for adults (0.7931). However, during middle meatal and other procedures (antral access or irrigation), children and adolescents routinely require general anesthesia or intravenous sedation [3]. Therefore, middle meatus samples were more specific (0.9600) in children compared with adults (under local anesthesia only) (0.6704). This may also explain the difference between the acute and chronic disease subgroups. Most samples were collected from patients with acute sinusitis in the outpatient clinic. In contrast, for those with chronic sinusitis, samples were often collected under general anesthesia prior to endoscopic sinus surgery. Thus, higher sensitivity (0.8219) and specificity (0.8226) were seen for the chronic than acute subgroup (0.7184 and 0.6694, respectively), as well as higher diagnostic accuracy (DOR = 18.2397 vs. 4.2660; *p* = 0.0373).

Recently, several studies using microbiome analysis for bacterial identification have been conducted [26,27,28,29], and literature has been reported that it is effective for the management of rhinosinusitis in children. Since specimens obtained from MMC in children have sufficient diagnostic value (especially suction aspiration) [11,13], performing microbiome analysis using them will provide useful information, particularly in differentiating children sinusitis management.

In terms of sample collection, the use of suction to collect mucous from a narrow nasal passage is associated with a lower risk of contamination than swabbing, because the suction sample is placed in a sterile container prior to withdrawal through the (contaminated) nasal vestibule [1,30]. This may help explain our results.

Our meta-analysis had some limitations. First, the number of studies analyzed was small, despite extensive literature searches. In addition, about two-thirds of all patients were from the USA or Taiwan, so the findings may not generalize to other regions or ethnicities. Second, although the maxillary sinus has traditionally been considered the major site of sinusitis, the osteomeatal complex and adjacent ethmoid sinuses are thought to be the initial sites. Therefore, antral tap may not provide reliable information on the microbiological status of other sinuses [17]. Third, none of the included studies used randomized designs and the assessors were typically unblinded. Therefore, it is likely some misclassification occurred, leading to a risk of bias. Nevertheless, our study provides useful information on the diagnostic utility of MMC compared with MSA/culture.

## 5. Conclusions

MMC showed fair diagnostic accuracy for patients with acute and chronic sinusitis. The accuracy was highest for patients with chronic sinusitis sampled via suction.

## Figures and Tables

**Figure 1 jcm-11-06069-f001:**
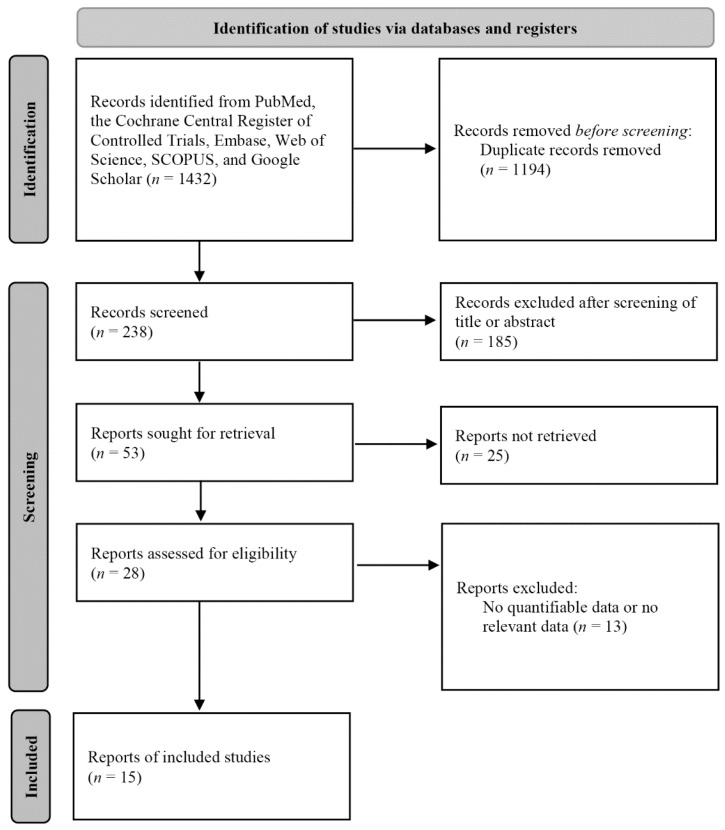
Summary of the search strategy.

**Figure 2 jcm-11-06069-f002:**
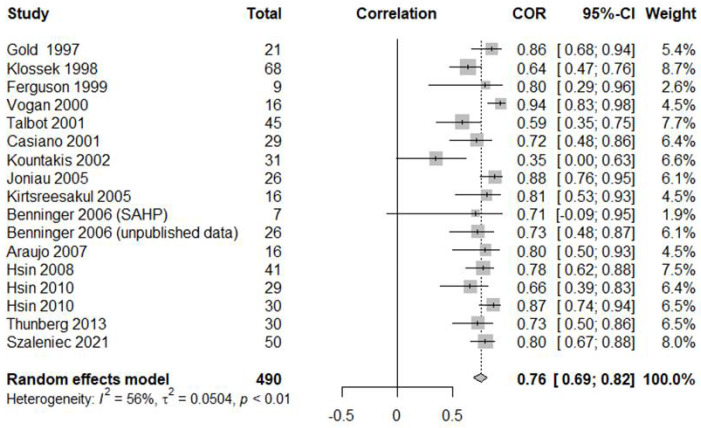
Correlation between the middle meatus and maxillary sinus culture results [1,3,4,6,7,8,9,10,11,12,13,17,18,19,20].

**Figure 3 jcm-11-06069-f003:**
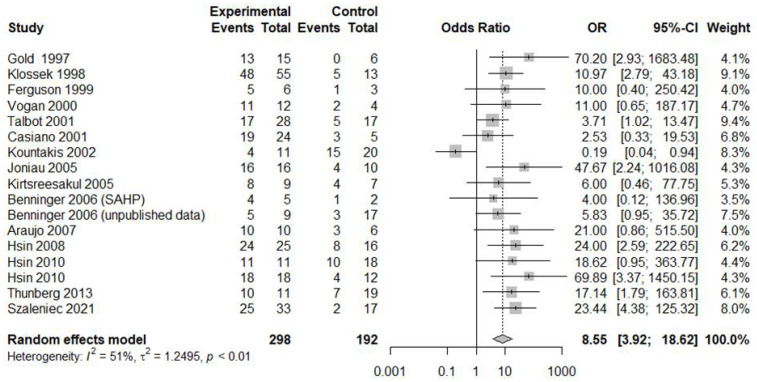
Forest plot of the diagnostic odds ratios of all studies [1,3,4,6,7,8,9,10,11,12,13,17,18,19,20].

**Figure 4 jcm-11-06069-f004:**
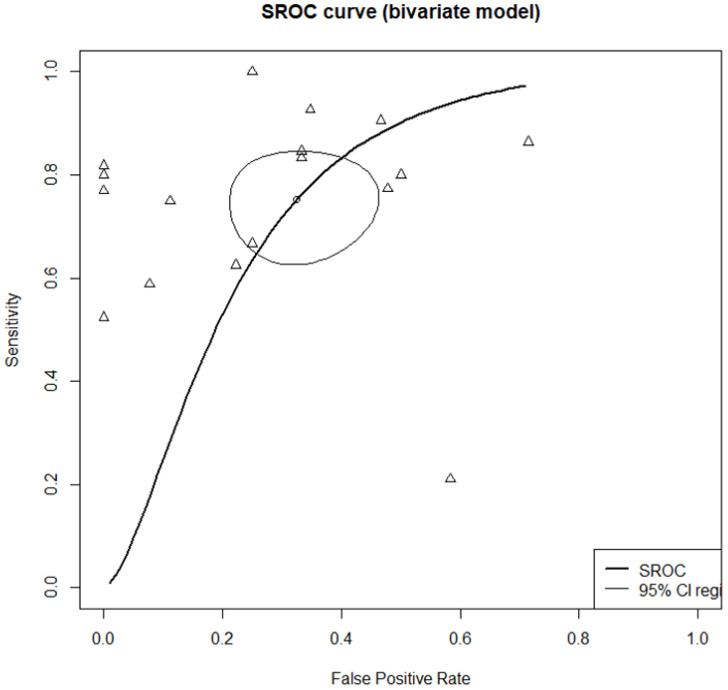
Areas under the summary receiver operating characteristic curves of all studies.

**Figure 5 jcm-11-06069-f005:**
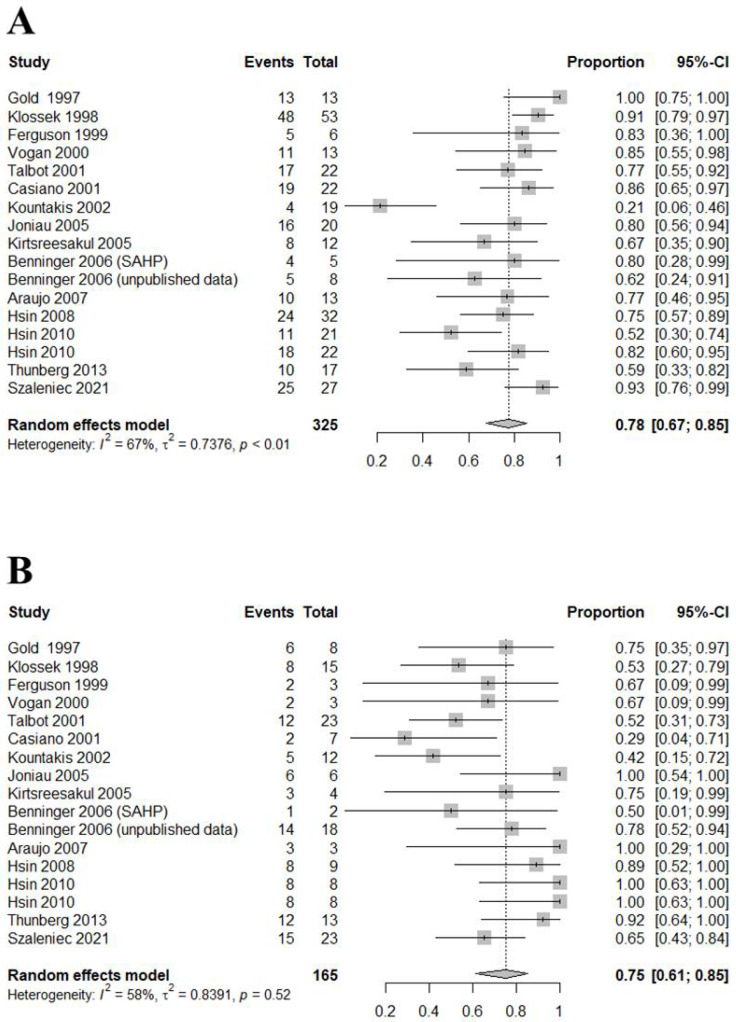
Forest plots of the sensitivities (**A**) and specificities (**B**) of all studies [1,3,4,6,7,8,9,10,11,12,13,17,18,19,20].

**Table 1 jcm-11-06069-t001:** Subgroup analysis by detection method.

	Sensitivity	Specificity	DOR	Correlation
Total (*n* = 17)	0.7759 [0.6744; 0.8526]; I^2^ = 67.4%	0.7514 [0.6110; 0.8534]: I^2^ = 58.2%	8.5475 [3.9238; 18.6199]; I^2^ = 51.0%	0.7613 [0.6917; 0.8169]; I^2^ = 56.0%
Adult (*n* = 14)	0.7931 [0.6734; 0.8769]; I^2^ = 68.0%	0.6704 [0.5461; 0.7747]; I^2^ = 38.5%	6.8726 [2.9181; 16.1863]; I^2^ = 54.1%	0.7563 [0.6716; 0.8215]; I^2^ = 58.7%
Children (*n* = 3)	0.7079 [0.5600; 0.8219]; I^2^ = 36.5%	0.9600 [0.7645; 0.9944]; I^2^ = 0.0%	29.5081 [6.3469; 137.1903]; I^2^ = 0.0%	0.7849 [0.6468; 0.8732]; I^2^ = 48.4%
	*p* = 0.3152	*p* = 0.0193	*p* = 0.1045	*p* = 0.6809
Acute sinusitis (*n* = 9)	0.7184 [0.5553; 0.8391]; I^2^ = 63.4%	0.6694 [0.4740; 0.8197]; I^2^ = 55.1%	4.2660 [1.3833; 13.1559]; I^2^ = 57.8%	0.7458 [0.6040; 0.8418]; I^2^ = 67.9%
Chronic sinusitis (*n* = 8)	0.8219 [0.7039; 0.8995]; I^2^ = 63.0%	0.8226 [0.6144; 0.9310]; I^2^ = 50.8%	18.2397 [8.3954; 39.6272]; I^2^ = 0.0%	0.7745 [0.7038; 0.8300]; I^2^ = 31.5%
	*p* = 0.2338	*p* = 0.2256	*p* = 0.0373	*p* = 0.6637
Biopsy (*n* = 1)	0.8636 [0.6521; 0.9554]; I^2^ = NA	0.2857 [0.0720; 0.6734]; I^2^ = NA	2.5333 [0.3286; 19.5311]; I^2^ = NA	0.7200 [0.5416; 0.8984]; I^2^ = NA
Suction (*n* = 7)	0.7979 [0.7047; 0.8672]; I^2^ = 0.0%	0.8400 [0.7114; 0.9179]: I^2^ = 0.0%	15.1597 [5.4793; 41.9424]; I^2^ = 0.0%	0.8353 [0.7734; 0.8814]; I^2^ = 0.0%
Swab (*n* = 9)	0.7386 [0.5637; 0.8607]; I^2^ = 78.5%	0.7000 [0.5256; 0.8308]: I^2^ = 56.2%	6.7136 [2.0918; 21.5475]; I^2^ = 67.9%	0.7115 [0.5944; 0.7990]; I^2^ = 65.0%
	*p* = 0.5355	*p* = 0.0158	*p* = 0.2553	*p* = 0.0547

DOR; diagnostic odds ratio, CI; confidence interval.

## Data Availability

The raw data of individual articles used in this meta-analysis are included in the main text or Appendix A.

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
