# Peer review of "Diagnostic Value of Middle Meatal Cultures versus Maxillary Sinus Culture in Acute and Chronic Sinusitis: A Systematic Review and Meta-Analysis"

_jcm, 2022, doi:10.3390/jcm11206069_

Round 1
Reviewer 1 Report
The Authors report the results of a systematic review and meta-analysis of middle metal colture to identify the etiology of acute and chronic sinusitis. It is a well detailed work but I think that the conclusion does not provide an advancement of the current knowledge.
I have some questions:
Middle meatus drains secretions not only from maxillary sinus but also from anterior ethmoid cells and frontal sinus: don't you think that using the maxillary sinus to verify the correctness of middle meatus sampling can be a source of bias?
The literature agrees that microbiome analysis could add useful information in particular in differentiating children sinusitis management: could you discuss this argument?
most of paranasal sinus surgery is performed for chronic sinusitis with nasal polyps: was considered the presence or absence of polyps in your meta-analysis? or only chronic sinusitis without nasal polyps was included?
lines 65 to 69: template instructions must be removed
Author Response
The pages and lines written in [Response to reviewers] are when "All Markup" is displayed with "Tracking Changes" feature enabled in the Word file.
The Authors report the results of a systematic review and meta-analysis of middle metal colture to identify the etiology of acute and chronic sinusitis. It is a well detailed work but I think that the conclusion does not provide an advancement of the current knowledge.
I have some questions:
Middle meatus drains secretions not only from maxillary sinus but also from anterior ethmoid cells and frontal sinus: don't you think that using the maxillary sinus to verify the correctness of middle meatus sampling can be a source of bias?
â—Ž Reply:
According to the reviewer's opinion, secretion in the maxillary sinus may include anterior ethmoid cells and frontal sinus. Therefore, the reason for this meta-analysis is to draw conclusions by collecting the papers reporting the culture results of middle meatal culture and maxillary sinus puncture. Overall, inferring the bacterial flora in maxillary sinusitis using middle meatal culture showed fair diagnostic accuracy, and the sampling method using suction from chronic sinusitis patients was the most accurate.
The literature agrees that microbiome analysis could add useful information in particular in differentiating children sinusitis management: could you discuss this argument?
â—Ž Reply:
According to the reviewer’s comment, we added the following text to the discussion section (Line 212 to 217):
Recently, a number of studies using microbiome analysis for bacterial identification have been conducted, and literature has been reported that it is effective for treatment of rhinosinusitis in children. Since specimens obtained from MMC in children have sufficient diagnostic value (especially suction aspiration), performing microbiome analysis using them will provide useful information in particular in differentiating children sinusitis management.
most of paranasal sinus surgery is performed for chronic sinusitis with nasal polyps: was considered the presence or absence of polyps in your meta-analysis? or only chronic sinusitis without nasal polyps was included?
â—Ž Reply:
Because the included studies did not mention polyp or were surgical treatment in the non-respondent to antibiotics drug therapy group, it was not possible to proceed with the analysis by dividing the presence of polyp.
lines 65 to 69: template instructions must be removed
â—Ž Reply:
We removed the template instructions from Line 67 to 69.

Reviewer 2 Report
The topic is relevant and interesting. The subject is not original, but is very important, and always contemporary. It adds a summary of the current knowledge in that area. The paper is well written, and the text is clear but not easy to read, because of a lot of statistics. The conclusions are consistent with the evidence presented, and they address the main question posed.
A minor correction is the inclusion EPOS 2020 document in the reference list. Also, I think the paper must be examined by specialists in medical statistics.
Author Response
The pages and lines written in [Response to reviewers] are when "All Markup" is displayed with "Tracking Changes" feature enabled in the Word file.
The topic is relevant and interesting. The subject is not original, but is very important, and always contemporary. It adds a summary of the current knowledge in that area. The paper is well written, and the text is clear but not easy to read, because of a lot of statistics. The conclusions are consistent with the evidence presented, and they address the main question posed.
A minor correction is the inclusion EPOS 2020 document in the reference list. Also, I think the paper must be examined by specialists in medical statistics.
â—Ž Reply:
According to the reviewer’s comment, we added EPOS 2020 as a reference (ref. 5).

Reviewer 3 Report
Dear authors,
Some errors of spelling and missing words could be detected. See Introduction, first chapter, describing the aspiration for maxillary sample. Also see Table S1, explanations of abreviations. Please check that all abbreviations in tables have been explained.
The meta-analysis increases current knowledge, and subgroup analysis is interesting finding good accuracy of MMC also in children. The number of patients is a problem especially in subgroups,, but conclusion are cautious. The analysis was carefully done. The manuscript is compact. The need for culture directed treatment is well justfied.
Author Response
Dear authors,
Some errors of spelling and missing words could be detected. See Introduction, first chapter, describing the aspiration for maxillary sample. Also see Table S1, explanations of abreviations. Please check that all abbreviations in tables have been explained.
â—Ž Reply:
We revise the phrases pointed out by reviewers in the introduction section.
We fixed the typo in Table S1 (Ture→True). all abbreviations in tables were explained below the Table.
The meta-analysis increases current knowledge, and subgroup analysis is interesting finding good accuracy of MMC also in children. The number of patients is a problem especially in subgroups,, but conclusion are cautious. The analysis was carefully done. The manuscript is compact. The need for culture directed treatment is well justfied.
â—Ž Reply:
We have removed from the conclusion section that is not inferred from the results of this analysis.

Round 2
Reviewer 1 Report
I'm still convinced that the study does not provide an advancement of the current knowledge